# Organophosphate Esters (OPEs) Flame Retardants in Water: A Review of Photocatalysis, Adsorption, and Biological Degradation

**DOI:** 10.3390/molecules28072983

**Published:** 2023-03-27

**Authors:** Yi Dang, Kexin Tang, Zhihao Wang, Haopeng Cui, Jianqiu Lei, Denghui Wang, Ning Liu, Xiaodong Zhang

**Affiliations:** 1School of Environment and Architecture, University of Shanghai for Science and Technology, Shanghai 200093, China; 2Shanghai Institute of Optics and Fine Mechanics, Chinese Academy of Sciences, Shanghai 201800, China

**Keywords:** organophosphate esters, photocatalysis, adsorption, biological methods, mechanism

## Abstract

As a substitute for banned brominated flame retardants (BFRs), the use of organophosphate esters (OPEs) increased year by year with the increase in industrial production and living demand. It was inevitable that OPEs would be discharged into wastewater in excess, which posed a great threat to the health of human beings and aquatic organisms. In the past few decades, people used various methods to remove refractory OPEs. This paper reviewed the photocatalysis method, the adsorption method with wide applicability, and the biological method mainly relying on enzymolysis and hydrolysis to degrade OPEs in water. All three of these methods had the advantages of high removal efficiency and environmental protection for various organic pollutants. The degradation efficiency of OPEs, degradation mechanisms, and conversion products of OPEs by three methods were discussed and summarized. Finally, the development prospects and challenges of OPEs’ degradation technology were discussed.

## 1. Introduction

With the development of human society and rapid industrial growth, organophosphate esters (OPEs) as flame retardants were used in the construction industry, automotive industry, and chemical industry to prevent fires. Owing to the irreversible pollution and toxicity of brominated flame retardants, BFRs had either been restricted or the research into them had been phased out over the years, and as a result, the production of alternative flame retardants such as OPEs had been increased. The global annual consumption of OPEs was 680,000 tons in 2015, and it is anticipated to reach 860,000 tons by 2023 [1,2]. Since no stable chemical bonds were formed between OPEs and other chemicals, they are usually released into the surrounding environment by physical means such as evaporation and wear [3,4]. The current studies have detected the presence of these substances in water, air, soil, fish, the human body, and other ecosystem environments. According to the different side chain structures, OPEs could be divided into three groups, including alkyl OPEs, halogenated OPEs, and aryl OPEs [5]. The stability of aryl groups halogenated OPEs was greater than that of halogenated OPEs and alkyl OPEs. At present, detection methods for OPEs are limited [6,7]. The concentration of OPEs was measured by liquid chromatography/tandem mass spectrometry (LC–MS/MS), which was equipped with electrospray ionization (ESI) and atmospheric pressure chemical ionization (APCI) [8]. According to Fauvelle et al. [9], 25 organic additives detected in deep water under the influence of hydrostatic pressure and prokaryotes were released into deep water along with plastic particles, including 9 OPEs, such as tripropyl phosphate (TPP); tri-iso-butyl phosphate (TiBP); tri-n-butyl phosphate (TnBP); tris (2-chloroethyl) phosphate (TCEP); tris-(2-chloro,1-methylethyl) phosphate (TCPP); tris-(2-chlor-,1-chloromethylethyl) phosphate (TDCP); triphenyl phosphate (TPhP); 2-ethylhexyl-diphenyl phosphate (EDHPP); tris-(2-ethylhexyl) phosphate (TEHP). TCEP, as one of the highest concentrations of analytes, was frequently detected in the water environment. Chen et al. [10] determined the presence of non-halogenated, chlorinated, and brominated organophosphorus flame retardants (TCPP, TCEP, and TBEP) in herring eggs by LC–MS/MS. In addition, OPEs were also detected in the air. Recycling e-waste emits TPhP and diphenyl phosphate (DPhP). The report showed that in an e-waste dismantling park and surrounding area in south China in the recent three years, tri-OPE concentrations and di-OPE concentrations in this area were much higher than in the surrounding area [11]. Belgium, Italy, Spain, and Nepal all had measurements of OPE concentration in household dust [12]. Meanwhile, the study also indirectly suggested that OPEs might enter the human body by some means by measuring the absorption of pollutants by adults and children.

With the increase of OPE emissions in water, many studies have focused on the harm in recent years [13,14]. Although triphosphate compounds had not been found to be bioaccumulative, certain monoesters and diesters of their degradation intermediates had been found to be endocrine disrupting and neurotoxic [15,16]. They had effects on water bodies, plants, aquatic organisms such as fish, and ultimately through the food chain or direct contact with the human body [17]. Of the halogenated OPEs studied so far, TCPP and TCEP were the most difficult to degrade and were prevalent in water environments [18,19]. TCEP, TPhP, and tris (1,3-dichloro-2-propyl) phosphate (TDCIPP) had been shown to cause spinal deformation in killifish and zebrafish larvae at concentrations of 300 μg/L [20], which indicated that OPEs had aquatic toxicity. In addition, Wang et al. [21] summarized human exposure to common types of OPEs. The levels of OPEs in the urine of adults were higher than those of infants, and firefighters had higher levels of OPEs than ordinary people, indicating that OPEs had affected humans and were related to human exposure pathways.

Based on the biotoxicity of OPEs, Figure 1 shows some key knowledge gaps regarding these reactive OPEs. Firstly, the impurity, migration, and leaching capacity of reactive OPEs in the products needed to be extensively investigated. Although these reactive OPEs could in principle form covalent bonds with materials such as polymers, it was not known whether there were unreacted OPEs in the products or whether they could migrate to the environment. Furthermore, the physical properties of OPEs showed that they were insoluble in water and aliphatic hydrocarbons but soluble in organic solvents, with a small amount of decomposition under alkaline conditions [22]. Therefore, it was worth noting that the degradation effect was affected by environmental factors. To better understand their health risks, these gaps must be addressed to find efficient, environmentally friendly, and thorough degradation methods.

Due to the wide application of OPEs, a large number of OPEs containing wastewater need to be treated every year, or the excess wastewater would be discharged into the water, posing a threat to the environment. Conventional wastewater treatment plants were not effective in removing OPEs [24,25]. In particular, chlorinated OPEs cannot be removed [26]. Furthermore, the P–O bond in OPEs was more stable to hydrolysis than the P–S and P–F bonds in most organophosphorus pesticides [27]. Therefore, it was necessary to develop a variety of clean, efficient, and promising treatment technologies to remove OPEs from aqueous matrices. The current treatment methods for OPEs could be divided into three categories: photocatalysis, adsorption, and biological treatment. The photocatalysis method was to stimulate the surface of the photocatalyst to produce strong oxidizing free radicals under light conditions. These free radicals attacked the OPEs’ structure for the purpose of degradation [28]. Metal-organic frameworks (MOFs) were a new material with a nanometer network, which were the most promising photocatalysts at present and commonly used catalysts for OPEs under visible light conditions [29,30,31]. Such as MIL-101 (Fe), MIL-88A, and MIL-88B-NH_2_. Compared with traditional photocatalysts, MOFs had the advantages of ultra-high specific surface area, high porosity, abundant unsaturated metal sites, and adjustable structure [32,33]. In order to improve the adsorption effect of one component adsorbent, a variety of methods or new adsorbents were selected [34]. The mechanisms of OPEs degradation by the adsorption method included van der Waals forces, electron donor–acceptor (EDA) interactions, hydrogen bonding, covalent bond formation, etc. Biological treatment was considered an environmentally friendly method that generally relied on esterase and was associated with cellular hydrolytic metabolism [35]. It had some disadvantages, however, such as low efficiency, a long degradation time, and a large footprint. Figure 2 shows a timeline of the development of technological breakthroughs in the removal of OPEs from water bodies. It could be seen that according to the chronological development, the first stage was the removal of OPEs by adsorption using carbon materials. The second stage was the removal of the hard-to-degrade chlorinated OPEs using photocatalytic materials and improved photocatalysis methods. In the third stage, in recent years, the researchers have focused on biological methods with green advantages and investigated degradation products that could be used in an adjustable way in the degradation process. These representative research methods were also categorized by technology and were involved in a later paper. In addition, there was also an increasing number of studies from 2020 to the present that tended to examine the concentration of OPEs in the river and lake waters in various regions of China as well as their biotoxicity [36,37].

At present, there are many aspects involved in the degradation methods of OPEs, but few studies have systematically summarized the various degradation methods and the commonalities of these methods, such as influencing factors or degradation products. In this paper, the photocatalysis method, the adsorption method, the biological method, and other degradation methods were reviewed. Additionally, the application of improved photocatalysis and adsorption methods to the degradation of OPEs was discussed. In addition, the degradation mechanisms of the three methods and the main degradation products of various OPEs were summarized. Finally, combined with the limitations of current OPE degradation methods, it was proposed that the new method combining multiple technologies should be studied in the future.

## 2. Photocatalysis Method

Photocatalysis technology uses photons to excite the semiconductor, making electrons in the semiconductor valence band (VB) transition to the conduction band (CB), and generating holes at the VB position. Photogenerated charge migrated to the surface of the semiconductor and was captured, and a series of chemical reactions occurred at the interface to degrade pollutants. The photocatalysis method had the advantages of high efficiency, greenness, low energy consumption, and a wide application range, so it was the most promising treatment technology at present [45,46]. However, the photocatalysis method required the addition of chemical reagents or the introduction of other toxic substances, and there was a risk of secondary pollution to the environment [47,48]. In the section on photocatalytic degradation of OPEs, this paper mainly introduced various MOFs and their modified materials and also included separate photodegradation. According to the light conditions, photocatalytic degradation of OPEs could be divided into UV, visible light conditions, and UV-vis, and the reaction could be in H_2_O_2_ or persulfate systems.

### 2.1. UV Conditions

Under UV conditions, TiO_2_ as a low-cost and environmentally friendly photocatalyst also showed great potential in the photocatalytic degradation of organic pollutants [49,50]. In particular, UV/TiO_2_ photocatalysis, one type of UV-AOPs in which an electron of a semiconductor was excited via light illumination, resulting in the formation of non-selective reactive species, had exhibited enormous potential for the elimination of recalcitrant organic contaminants. Yu et al. [38] proved that UV/TiO_2_ multiphase photocatalysis was an effective method to remove TCPP; with 1 mg/L TCPP in the UV/TiO_2_ system, the degradation could reach 95.1% in 10 min. However, there was no degradation efficiency of OPEs driven by UV irradiation alone. This indicated that the presence of a photocatalyst was effective in degradation. The quenching experiment proved that •OH radicals played a dominant role in photocatalytic reactions, and the free radical attack led to the formation of three intermediates. TCPP was composed of C, H, O, Cl, and P, so the final degradation products of TCPP contained CO_2_, H^+^, Cl^−^, and PO_4_^3−^. The reaction equation was Equation (1). The UV/H_2_O_2_ process was a conventional advanced oxidation process based on the photolysis of H_2_O_2_ to produce the hydroxyl radicals (•OH). According to He et al. [51], the final degradation experiment results showed that the yields of Cl^−^ and PO_4_^3−^ reached 0.19 mg L^−1^ and 0.58 mg L^−1^, respectively, and the TOC removal rate was 43.02%. The degradation rate of 5 mg L^−1^ TCPP in the UV/H_2_O_2_ system could reach 96% within 15 h. This was due to the fact that, in advanced oxidation, UV light contributed to the generation of reactive radicals that helped to degrade OPEs.
(1)$$C_{9}H_{18}{\mathrm{Cl}}_{3}{\mathrm{PO}}_{4}\overset{\mathrm{Oxidation}}{\to }\;{\mathrm{CO}}_{2}+H^{+}+{\mathrm{Cl}}^{-}\;+\;{\mathrm{PO}}_{4}{}^{3-}+\mathrm{Intermediates}$$

In addition, studies had shown that the concentration of pH, inorganic anion, and humic acid (HA) had an effect on the degradation rate of TCPP [52]. Among these UV-AOPs, pH was considered to be a vital factor affecting light reactions. The results showed that strong acids or bases were not conducive to TCPP degradation, and the most suitable pH range was 5–9 [53]. Chen et al. [54] used 185 + 254 nm UV irradiation to degrade TCEP. The emission of UV could induce the homolysis and ionization of water and other simple molecules to generate •OH. This study confirmed that irradiation photolysis at 185 nm and 254 nm and the oxidation of •OH could simultaneously promote the degradation of TCEP. Quantitative analysis by screening the three mechanisms revealed that although 185 nm UV alone could degrade TCEP, the photolytic efficiency was not significant and could be ignored. The quenching experiment confirmed that the oxidation of •OH played a major role in UV treatment. Similarly, the optimum pH for TCEP degradation should be kept between 5 and 7, and the degradation efficiency would be severely inhibited under the strong acid and strong base. In practical water treatment, the pH was usually kept at a neutral state, which facilitated TCEP degradation.

### 2.2. Visible Light Conditions

As a kind of porous material with a customizable structure and adsorptive properties, MOFs had better application potential in photocatalytic reduction and photocatalytic degradation of organic pollutants [55,56]. Under visible light conditions, the degradation of OPEs by MOF photocatalysts in the PMS system and Fenton system was mainly introduced. Hu et al. [57] prepared MIL-101 (Fe) and used the MOF/photo/persulfate system to degrade TCEP. The reaction process consisted of reactant adsorption and photocatalyst activation. On the one hand, MOFs used visible light activation to convert Fe into Fe^2+^ and promoted the conversion of S_2_O_8_^2−^ in the persulfate system into SO_4_^•−^. On the other hand, as with the catalytic oxidation mechanism, TCEP was oxidized and degraded into various products with low toxicity by free radicals or eventually mineralized into Cl^−^ and PO_4_^3−^. By comparing the degradation of TCEP by irradiation of different wavelengths, it was found that short-wavelength visible light could also induce a photocatalytic reaction in the system, but the total degradation effect was 420 > 280 > 310 > 472 nm. A special S-shaped degradation curve was observed at 420 nm. In the study of pH, it was found that the MOF/photo/persulfate system had the best degradation efficiency under acidic conditions (pH = 3). This result was different from the pH effect of light reaction alone. The reason might be that there was more H^+^ in the reaction system under acidic conditions, and the protonation reaction induced a positive charge on the surface. As a result, more persulfate (S_2_O_8_^2−^) was adsorbed by Fe-MOF, which promoted reactant transport and shortened the time to induce a reaction. Liu et al. [58] synthesized Fe-MOF (MIL-88A) as a photo-Fenton catalyst to degrade TCPP under visible light conditions. In the H_2_O_2_/Vis system, TCPP degradation was weak. The degradation rate of the MIL-88A/H_2_O_2_/Vis system to TCPP was about 95%. It might be due to the Fe–O clusters in the MIL-88A activating H_2_O_2_ and consequently forming the active •OH. With the development of MOF materials, it was found that the application of the original photocatalytic materials could be improved by the construction of heterogeneous junctions or the formation of defects [59,60]. Liu et al. [61] went on to synthesize amino-functionalized MOFs (MIL-88B-NH_2_), which were used as catalysts for TCPP degradation in the H_2_O_2_/Vis system. The kinetic constant of TCPP degradation in the MIL-88B-NH_2_/H_2_O_2_/Vis system was 0.086 min^−1^, which was significantly higher than that in the MIL-88B/H_2_O_2_/Vis system (0.021 min^−1^). This was due to the dual excitation of Fe-O clusters and the fact that H_2_-ATA accelerates the transition from Fe^3+^ to Fe^2+^, resulting in the reaction with H_2_O_2_ to generate abundant •OH. The effect of pH on degradation efficiency was studied, and it was found that TCPP degradation efficiency could reach its highest within 30 min when pH = 3. The time required to reach the highest degradation efficiency increases with increasing pH. When the pH was from 3 to 7, TCPP was completely degraded within 60 min, which extended the pH range compared to the Fenton system without photocatalysis alone. At the same time, the pH effect range was also expanded compared to the PMS system. Lin et al. [62] modified MIL-101(Fe) by loading graphene oxide (GO) to prepare GO@MIL-101(Fe) and established a light/MOF/H_2_O_2_ photocatalytic system for TCEP degradation. Through comparing degradation experiments, it was found that under light/H_2_O_2_ treatment, only 280 nm/H_2_O_2_ could degrade TCEP. This was due to the UV-induced cleavage of H_2_O_2_ into •OH and the degradation of TCEP. However, both single-light source irradiation and light/catalyst had no effect on TCEP degradation. This indicated the importance of the H_2_O_2_ system on the generation of •OH induced by the MOF’s photocatalytic system.

The reaction equations for the two systems under light conditions were as follows (Equations (2)–(4)), where M represented MOF photocatalysts [63]: (2)$$\equiv \;M^{\left(n+1\right)}\overset{\mathrm{irradiation}/\mathrm{electron}\;\mathrm{transport}}{\to }\;\equiv \;M^{n+}$$
(3)$$\equiv \;M^{n+}+H_{2}O_{2}\;\to \;\equiv \;M^{(n+1)+}+{\mathrm{OH}}^{-}+•\mathrm{OH}$$
(4)$$\equiv \;M^{n+}+S_{2}O_{8}^{2-}\;\to \;\equiv \;M^{(n+1)+}+{\mathrm{SO}}_{4}^{2-}+{\mathrm{SO}}_{4}^{•-}$$

In addition, in the advanced oxidation methods, sulfate-free radical S-AOPs produced SO_4_^•−^ with a stronger oxidation potential (2.6 V) in a wider pH range and produced fewer toxic by-products than the •OH [64,65]. This was in contrast to the above conclusion that the pH influence range of the H_2_O_2_ system was larger than that of the PMS system in the photocatalysis method under visible light. This also indicated that the degradation efficiency depended not only on the oxidation capacity of free radicals but also on their activation ability. 

### 2.3. UV-Vis Conditions

UV-vis conditions simulated sunlight with wavelengths ranging from 220 to 800 nm. For the same photocatalyst, the degradation efficiency of UV-vis conditions was lower than that of pure UV conditions. Therefore, in addition to adding activators, the catalytic efficiency could be increased by doping modification. Doping consisted of metal doping and non-metal doping by forming impurity levels at low and high band gaps in semiconductors, respectively. These could not only improve the photoresponse range of the photocatalyst but also improve the mobility of photogenerated electrons so as to improve the photocatalytic performance [66,67]. For example, Antonopoulou et al. [39] prepared N and N, S co-doped TiO_2_ catalysts via a simple sol-gel method to evaluate the degradation of TCPP in aqueous solutions under UV-vis light irradiation. The degradation results showed that the four different amounts of doped catalysts had better degradation efficiency than pure TiO_2_. On the one hand, there were surface acid sites on the N- and S-doped catalyst, which acted as the adsorption centers of TCPP and O_2_, and the adsorption performance of the catalysts was improved. On the other hand, increasing the adsorption of O_2_ enhanced electron capture, leading to efficient charge separation and the further formation of O_2_^−^ and •OH. Under visible light conditions, it was mainly reactive oxygen species that played a role in TCPP. The degradation kinetics of TCPP under visible light irradiation were slower than those under UV-Vis irradiation; this might explain the reduced production of •OH under visible light irradiation. Metal doping refers to the doping of transition metals or heavy metals such as Pt, Pd, Au, and Ag in the material to improve the degradation efficiency of the photocatalytic reaction [68]. Panagiotis et al. [69] used N- and F-doped TiO_2_/V_2_O_5_ and N-doped SrTiO_3_ as visible light-responsive catalysts for multiphase photocatalysis under simulated sunlight to remove the TBEP from water. V_2_O_5_ is a non-ferrous metal. Firstly, TiO_2_/V_2_O_5_ composite material had shown good degradation efficiency, which increased to more than 95% after mixing N and F, while the degradation efficiency of N-doped SrTiO_3_ material could reach 100%. Analysis of degradation pathways showed that •OH acted as the major oxidant during degradation, which was achieved gradually through several hydroxylation and oxidation steps, leading to the dealkylation of TBEP. UV-vis conditions were consistent with the main mechanism of degradation of OPEs by UV or visible light alone, in which •OH was the main active substance produced by photoexcitation. The photocatalytic effect of irradiation on OPEs was UV > UV-vis > visible light. It also indicated that doping modifications had a significant effect on improving degradation efficiency. The methods and efficiency of photocatalytic degradation of OPEs were summarized in Table 1. The key to the photocatalysis method was how to improve the photoresponse range of the catalyst and carrier migration ability.

## 3. Adsorption Method

OPEs with different substituents had different molecular sizes and shapes, which subsequently affected their sorption affinity and capacity. In the adsorption method, the substituents played important roles in sorption processes due to EDA interactions, hydrogen bonding, and covalent bond formation mechanisms. Table 2 summarizes the studies on the degradation of OPEs by adsorption.

### 3.1. Carbon Materials

Carbon-based adsorbent materials included three categories: nanoadsorbents, such as graphene, GO, and carbon nanotubes; biomass-based materials that could be used or recycled or modified with MOFs; MOFs synthesized by composite nanotechnology and activated carbon (AC); or composite materials synthesized by carbon nanotubes or graphene. AC as a traditional carbon adsorbent was dominated by a microporous structure with functional groups on the surface of C-O, C=O, O-H, and C=C. These oxygen-containing functional groups provided binding sites for the chemisorption of organic pollutants. Traditional adsorbents had the advantage of good adsorption performance for a variety of substances, but their structure was complex and porous, and they had low selectivity. Thus, various modern carbon adsorption materials had been developed: carbon nanotubes, graphene, and biochar. Carbon nanotubes were advanced adsorbents designed on the basis of AC. Biosorbents are referred to as biochar, which is the residue produced by the hydrothermal carbonization of biomass. Compared to AC, biosorbents were attractive due to their renewable ability and cost-effectiveness. Unlike AC, biochar did not require physical or chemical activation in the preparation process and used renewable biological feedstocks [74,75].

Wang et al. [76] selected five OPEs as model pollutants for the first time to evaluate the adsorption behavior of different OPEs on AC. Four different sizes of AC were selected: granular activated carbon (GAC), powdered activated carbon (PAC), reactivated activated carbon (R-GAC), and oxidized activated carbon (O-GAC). Among them, PAC had more adsorption sites for OPEs, a faster adsorption rate, and greater adsorption of OPEs, and the effect was significantly higher than GAC. This indicated that the size of the AC had a great influence on the adsorption of OPEs. Different types of OPEs have different hydrophobic properties [77]. In the environment, monoesters and diesters existed as binary ions or single anions in a dissociated state; they were more hydrophobic, and their adsorption of particles and accumulation in biota would be weaker [78,79]. The adsorption kinetics of OPEs conformed to the pseudo-second-order model, with the initial adsorption rate (V_0_) of OPEs on PAC being TEP < TCPP < TCEP < TBP < TPhP, which was consistent with their hydrophobicity. Since the adsorption process of porous adsorbents mainly depends on the internal diffusion of particles, the molecular size of OPEs also affects the adsorption effect. The TCPP structure had complex aliphatic substituents, which might break the P–O bond to form hydrogen bonds with -OH and -COOH on the surface of AC, resulting in a lower initial adsorption rate than TCEP [52]. The initial adsorption rates of aromatic OPEs on PAC and GAC were significantly higher than aliphatic OPEs. According to the properties of the five OPEs substrates, different adsorption mechanisms included hydrophobic interaction, π-π interaction, electrostatic attraction, and hydrogen bonding. Yan et al. [40] explored the adsorption process and molecular mechanism of OPEs on multi-walled CNTs (MWC-NTs), single-walled CNTs (SWCNTs), and their oxidative counterparts (O-MWCNTS and O-SWCNTs) in a wide range of concentrations. Carbon nanotubes are carbon nanoparticles with unique physical and chemical properties. The oxygen-containing groups on the surface were mainly carbonyl, hydroxyl, ester, and carboxyl groups, which could provide an abundance of adsorption sites for the adsorption and aggregation of organic matter and affect the adsorption capacity of organic matter. The study also confirmed that the adsorption affinity of OPEs was related to hydrophobicity, which indicated that van der Waals forces dominated the carbon material for OPEs. Fitting to the Dubinin–Ashtakhov adsorption isotherm model suggested that the oxygen groups indicated by carbon nanotubes could form hydrogen bonds with water molecules. Raman and FT-IR spectroscopy confirmed that the EDA interaction also played a crucial role in the adsorption of aromatic OPEs. Three kinds of adsorption were the main mechanisms of OPEs adsorption on carbon nanotubes.

### 3.2. Carbon—Metal Composites

Ball milling technology was an efficient and environmentally friendly method for preparing nanostructured materials, which facilitated the uniform distribution of the adsorbent in water and enhanced the adsorption performance. Wang et al. [71] prepared a magnetic powder carbon adsorbent (PC/nano-Fe_3_O_4_ composite) by ball milling and used PC adsorption and Fe_3_O_4_ nanoparticle catalysis to remove TCPP. The material preparation method was simple, but it had a good adsorption effect. The maximum adsorption capacity was 2682.1 μg/g. TCPP could be completely degraded within 3 h in a Fenton-like reaction system, and about 90% of TCPP was removed within 8 h in the PMS system. In addition, it was found that only Fe_3_O_4_ loaded on PC had an effect on TCPP degradation; this was due to the carbon skeleton in PC promoting the electron transfer between TCPP and Fe_3_O_4_ nanoparticles. In the Fenton-like reaction system, low pH promoted TCPP degradation, while in the PMS activation system, high pH promoted TCPP degradation.

Graphene had a layered structure and could be doped with functional groups on the surface. It was also the most effective and commonly used carbon adsorbent. Metal or metal oxide doped graphene was attractive to improve the adsorption selectivity [70]. Chen et al. [80] prepared a 3D graphene-La_2_O_3_ composite as a 3D adsorbent for phosphate adsorption and found that the acidic conditions were conducive to phosphate adsorption. According to the variation in pH value in the reaction system, it could be deduced that the surface of the composite with a positive charge under acidic conditions had an electrostatic attraction with the phosphate with a negative charge. Compared with pure graphene, the composites doped with La_2_O_3_ not only had more than 90% adsorption capacity but also had a good tolerance to the presence of anions in water. The adsorption mechanism between phosphate and adsorbent studied by Wu et al. [81] was divided into hydrogen bonding, shape complementation, and inner sphere composite. There was also anion competitive selective adsorption, which also had a significant impact on the reusability of adsorbent-desorption. It could be seen that the composite-modified carbon-metal materials could improve the degradation of OPEs, and pH was still the main factor affecting the adsorption. The main adsorption mechanisms involved include electrostatic adsorption related to charge distribution, intraparticle diffusion, and hydrogen bonding. In addition, the adsorption effect of carbon-metal materials on different OPEs in different systems was also worth exploring.

### 3.3. Other Adsorbents

The adsorption capacity of these materials was low due to the P–O bond in OPEs [82]. Therefore, Wang et al. [72] chose widely used resin materials and studied for the first time the adsorption kinetics and isotherms of XAD4 hydrophobic and XAD7hP hydrophilic resin on five typical OPEs and the influence of pH, common influencing factors of water, adsorption mechanisms, recyclability, reuse, etc. The results showed that the adsorption kinetics and adsorption isotherm fitting results of the five OPEs differed due to the different structures and hydrophobicities of aliphatic and aromatic OPEs. On the one hand, influenced by the physical properties of OPEs, aliphatic OPEs had a small molecular space and were easy to diffuse between particles, which would reach adsorption equilibrium faster than aromatic OPEs. The maximum adsorption capacity increased with the hydrophobicity of OPEs increasing, which was reflected in the fitting of the adsorption isothermal model. On the other hand, it was related to the properties of the adsorbent. XAD4 had a higher specific surface area, so the adsorption capacity was stronger. Compared with similar hydrophobic aliphatic OPEs, the adsorption capacity of XAD7hP for two aromatic OPEs was lower than that of XAD4 with the structure of divinylbenzene, indicating that the possible π-π interaction existed in the adsorption process of aryl-OPEs on XAD4 resin. Based on the adsorption behavior of OPEs on the two resins, the hydrophobic, electrostatic repulsion, hydrogen-bridging, and π-π interactions were proposed in the adsorption process. The coexistence of anions and HA substances in the actual wastewater had an important influence on the removal of OPEs by XAD4, while the relatively hydrophilic XAD7hP had no significant influence on the removal of OPEs. This was due to the high concentration of affected substances that would occupy the adsorption site of XAD4 and the π-π interaction, resulting in pore blocking, while XAD7hP resin was less affected due to hydrophobicity. Although the above resin adsorbents could be used in three cycles in order to further improve the adsorption and degradation performances of resin adsorbents. Liu et al. [41] used iron oxide hydrate (HD1) resin-based nanocomposites as Fenton catalysts to effectively catalyze the decomposition and degradation of TCEP by hydrogen peroxide. The results showed that the prepared HD1 had good applicability, and catalytic and adsorption performance. The complete degradation of TCEP was achieved by combining the adsorption properties of resin-based nanomaterials with the catalytic oxidation of the Fenton system. The resin had excellent mechanical strength and an adjustable pore structure, so the HD1 nanocomposite adsorbent was synthesized by loading iron hydroxide (HFO) with high selectivity as a carrier. The synthesized HD1 was used as a heterogeneous Fenton catalyst to immobilize inorganic phosphorus (IP) through electrostatic interaction and inner layer complexation. Finally, the •OH generated by the Fenton system was mainly used for the degradation of TCEP.

In recent years, there have been other methods to further study the adsorption of OPEs. Microplastics that accumulated in seawater had a high specific surface area and could aggregate pollutants in water through adsorption, and the accumulation of OPEs also damaged the water environment. OPEs absorbed by microplastics in the ocean were considered a threat to the global marine environment [83]. Therefore, Chen et al. [73] selected polyethylene (PE) and polyvinyl chloride (PVC) microplastics as representative microplastics and used TnBP and TCEP as typical non-chlorinated and chlorinated OPEs to study the adsorption behavior of microplastics on OPEs. For PE, the adsorption effect of TnBP, which also had hydrophobicity, was higher than that of TCEP due to the hydrophobic effect of the adsorbent. On the contrary, PVC had a higher adsorption effect on TCEP, which was attributed to the fact that PVC was a chlorinated polymer and the hydrogen atom in its molecular structural unit was replaced by a chlorine atom, which gave PVC a heterogeneous composition in its molecular structure and a higher polarity than PE. Through polar-polar interaction, the affinity of the chlorine atom adsorption domain in PVC for TCEP was greater than TnBP, and the polarity and water solubility of TCEP were higher than those of TnBP. According to the fitting of the adsorption isotherm model, the adsorption of TnBP and TCEP on PE microplastics conformed to the Langmuir model, and the Freundlich isotherm could better fit the adsorption isotherm data of TnBP and TCEP on PVC microplastics. This indicated that the main mechanism of TnBP and TCEP adsorption by polyethylene microplastics was simple monolayer adsorption van der Waals forces, while the main mechanism of PVC microplastics adsorption on OPEs was a pore-filling interaction mechanism. Therefore, the adsorption behavior of microplastics on pollutants was complex, and the physical and chemical properties of different pollutants and microplastics had a significant impact on the adsorption behavior.

## 4. Biological Methods and Other Methods

### 4.1. Biological Enzymolysis

OPEs were widely used as plasticizers and flame retardants. Due to their hydrophobicity and biological accumulation, OPEs could accumulate in the environment and spread through the air and food chain. At present, concentrations of OPEs have been found in fish and some mammals in North America, Europe, and Asia. OPEs were found to accumulate mainly in fat and muscle, while the accumulation of OPEs in other tissues was minimal. In addition, the concentration of OPEs entering the organism gradually decreased with the accumulation of time due to the metabolic effects of the organism. Therefore, it was speculated that the biodegradation process of OPEs was closely related to the metabolic effect of cells [20,37]. Choi et al. [42] took TPhP as the research object and based their analysis on the producer-consumer-decompositionist relationship in the ecosystem. Microalgae production (Raphidocelissubcapitata), primary consumption invertebrates (Daphniamagna), and secondary consumption fish (Oryziaslatipes) were selected as experimental species, and trace elements and feed were added to simulate the aquatic ecological environment. Finally, the biodegradation products were analyzed, and the toxicity of the biotransformed information products was evaluated according to the prediction model. The results indicated that the biotransformation products of TPhP in different organisms were different, and the toxicity of the 29 products decreased after degradation. Based on this study, the source path of BTPs under different conditions of TPhP in the environment could be predicted.

Most OPEs had relatively stable haloalkyl or aryl ester bonds and degraded difficulty in the environment [84]. Enzymatic hydrolysis was a green technique to neutralize these refractory organic matters. Unlike the naturally occurring enzymes sb-PTE that could degrade OPEs, Zhang et al. [85] achieved 99% degradation of TPhP in 24 h and complete degradation in 48 h by introducing two mutant gene synthetic variants of PoOPHv5. The high hydrolysis activity of the biocatalyst was attributed to the synergistic effect of the mutation, which increased the hydrophobicity of the substrate binding bag and broadened the exit channel of the product. Based on the metabolism of organisms and enzymes in cells, Qin et al. [86] established three types of constructed wetlands (biofilm attachment surface CWS, packed bed CWS, and traditional CWS) to compare the long-term removal of TCPP in continuous flow. In this constructed wetland system, igneous rock filler and aquatic plant alternanthera were added to the filling device. The adsorption of phosphate by igneous rock as filler matrix was mainly surface adsorption and chemical precipitation and adsorption of CaO, MgO, or iron oxides. This was also the main reason for the high removal rate of TCPP in the early stage of CWs. Alternanthera alterniflora had a strong absorption capacity for pollutants, and its leaves could tolerate antibiotic pollution, which induced the secretion of more antioxidant enzymes by up-regulating the expression of proteins related to antioxidant defense and stress. The results showed that the removal rate (26–28%) of TCPP in two types of CWs containing plants was twice as high as that in plant-free CWs in the 6-month experiments, and this showed that aquatic plants played a role in the process of removing TCPP. However, TCPP would release from the packing after a period of time. In addition, it was found that TCPP transferred and migrated in hydrophytes, showing terminal accumulation, which may have demonstrated that TCPP could be transported and utilized via cellular metabolism [87]. Based on the proteomic analysis, Massilia, SM1A02, and Denitratisoma might be associated with the degradation of TCPP, and other bacterial genera might also be capable of degrading contaminants after long-term domestication. In addition, cell hydrolases facilitated the cleavage of C-C, C-O, C-N, and other bonds by water. Cell hydrolases also catalyzed several related reactions, including condensation and alcoholysis. The main advantages of this enzyme class were its availability, lack of cofactor stereoselectivity, and ability to tolerate the addition of water-miscible solvents [88]. It could be inferred that the increased metabolism and secretion of these proteins may enhance TCPP biodegradation in CWs.

### 4.2. Other Methods

Kim et al. [89] found that among 14 OPEs, tris(methylphenyl) phosphate (TMPP), tris(2-butoxyethyl) phosphate (TBOEP), and TEHP had removal efficiencies of slightly above 60% in water at a WWTP. However, the removal efficiencies of TPhP and bis (1,3-dichloro-2-propyl) phosphate (BDCIPP) were <40% to negative values, indicating that although OPEs had the removal efficiency in WWTPs, the removal was incomplete. Considering that treatment processes in WWTPs were not effective and the degradation of OPEs was seriously hindered by the quenching of various inorganic anions in actual wastewater. The bioelectrochemical system (BES) was a more efficient and sustainable technology that used domesticated electrochemical biofilms as biocatalysts to facilitate the conversion of energy from organic matter’s chemical energy to electrical energy [90]. Hou et al. [91] studied for the first time the feasibility of the degradation of TPhP and several non-chlorination OPEs by the BES system after a certain adaptation period. The results showed that at the optimal voltage of 0.2 V, the degradation time required for the open-loop group was 72 h, and the degradation efficiency of BES for TPhP was 1.44 times. The best degradation rate of TPhP by BES was at 1.0 g/L sodium acetate concentration. In this process, the acceded microbial community could use electrodes as electron acceptors to improve the degradation efficiency of TPhP and then enhance the removal of toxicity. A microelectric field and the addition of acetate as a co-substrate could promote the bacterial metabolism of TPhP. OPEs usually contained halogenated side chains, which were more difficult to remove than alkane OPEs and easily produced toxic by-products when degraded. A.M. et al. [92] reported the degradation of TCEP in an oxygen-saturated aqueous solution under ultrasonic irradiation and showed that ultrasonic dissolution treatment at 640 kHz resulted in rapid degradation of TCEP in an oxygen-containing aqueous solution with a mineralization yield of approximately 48% chloride and 32% phosphate. The mechanism of ultrasonic irradiation was to form micro-bubbles in violent motion by increasing the pressure around the water, and then the bubbles collapse rapidly and cavitation occurs when the temperature and pressure reach a certain degree. In this process, the chemical transformation of organic matter in water occurs, and the chemical bonds in water vapor are split into •H and •OH and H_2_O_2_ active substances for water treatment. Jose R. et al. [93] studied the degradation efficiency of TCEP by ultrasound (US) alone and in the presence of Na_2_S_2_O_8_, H_2_O_2_, O_2,_ and other reagents. The results showed that US and Na_2_S_2_O_8_ had significant synergistic effects, and the degradation rate constants of TCEP were 0.0068 min^−1^. However, these US methods’ degradation rates were lower than those of other alternative methods, such as UV/TiO_2_, UV/H_2_O_2_, O_3_/H_2_O_2_, and Fenton and Fenton-like processes. It could be seen that there were various treatment methods for OPEs. According to different substrate properties, efficient and environmentally friendly treatment processes could be selected, and on the basis of the original process, oxidants could be added to cooperatively produce free radicals to achieve the purpose of degradation.

## 5. Degradation Mechanism and Products

### 5.1. Photocatalysis Degradation Mechanism and Products

The mechanism of photocatalytic degradation of organic pollutants showed that the catalyst absorbed appropriate frequencies within a certain wavelength range of the UV radiation or light; the catalyst was greater than the width of the band gap energy of the photons illuminates; the electronic (e^−^) in the VB of the photocatalyst would be excited, and then from VB it would jump to the CB, form a highly active e^-^ on the CB, and form the h^+^ in the forbidden zone. The optical excitation of h^+^ obtained in electronic forms after its ability was improved could capture the catalyst surface adhesion of organic solvent or e^−^ and make the material that originally not absorb incident light with activated oxidation. The e^−^ acceptor energy was reduced by accepting electrons from the surface [94,95]. In a water solution, photocatalytic oxidation occurs on the surface of the catalyst; the main electron loss was water, and water molecules combined with holes could form •OH with strong oxidation ability [28]. In addition, because some OPEs side-chain alkyl groups lacked chromophores and did not absorb wave peaks, they could not be directly photodecomposed under UV and solar irradiation. Therefore, oxidants such as H_2_O_2_, TiO_2_, persulfate, and HA were often added to improve the production of active substances in improvement experiments. In the PMS or Fenton system, persulfate and hydrogen peroxide react with photocatalyzed electron holes to generate strong oxidizing SO_4_^•−^ and •OH to further oxidize organic compounds. The degradation of OPEs was mainly caused by the reaction of free radicals produced by photocatalysis. Studies had shown that the molecular structure of the branched chain of OPEs made the degradation rate of these substances follow a certain rule when they were degraded by the •OH radical [96]. For example, Antonopoulou et al. [43] first simulated, under the simulated sunlight, the TiO_2_ photocatalytic removal of TCPP in ultrapure water (UW) and actual wastewater (WW), and determined the main TPs. The results showed that the degradation was mainly conducted by •OH radical attacks. Moreover, the degradation of wastewater was significantly hindered by the inhibition of inorganic ions. UPLC-TOF-MS analysis showed that the main degradation pathways of TCPP were hydroxylation and oxidation, followed by dechlorination and dealkylation. The toxicity analysis showed that the toxicity of the solution would decrease after degradation treatment.

OPEs had different hydrophobicity due to their different side chain structures, and thus their stability in water was different. Their degradation degree was consistent with their stability. The order was that the aryl-OPEs > halogenated-OPEs > alkyl-OPEs; TCPP was a halogenated-OPE; and the •OH generated by photocatalysis in the degradation process was the easiest to attack the chlorine atom of the side chain. The degradation of alkane OPEs, such as TBEP, was mainly through dealkylation. •OH radicals extracted hydrogen from alkyl chains, first to form a carbon-centric radical, which was subsequently added by oxygen to form a peroxyradical. Then it was decomposed into hydroxylated and carbonylated products by the bimolecular Russell mechanism. The presence of O atoms in the TBEP aliphatic chain promoted the response to •OH [97,98].

The reactivity of the SO_4_^•−^ group was lower than that of the •OH group [99]. Taking TCEP as the target object, the process of PMS system photodegradation could be summarized [100] as shown in Figure 3: a. SO_4_^•−^ attacked the P atom, then removed the branched chain—CH_2_Cl to produce P1; b. SO_4_^•−^ attacked the C atom at the end of the branch chain to remove the Cl atom, and finally hydrolyzed to form P4. Furthermore, Wu et al. [101] detected TCEP products degraded by UV/H_2_O_2_ by Fourier transform ion cyclotron resonance mass spectrometry and deduced two possible degradation paths: a. •OH addition to P, breaking the P–O bond, and then breaking the branched chain; b. •OH attacking the C atom at the end of the branch chain to form the carbon-centered radical, followed by oxidation, the Russel reaction, the hydrolysis reaction, and finally P2 and P3. In addition, when •OH oxidized such substances, the first irreversible reaction mainly took place in the extraction of H from the alkyl group on the side chain or the addition of •OH to the P.

In general, the influencing factors of photocatalytic degradation of OPEs include light conditions, substrate structure, photocatalyst activity, oxidant activity, pH, inorganic anions, etc. [102]. On the one hand, acidic conditions were not conducive to the degradation of OPEs. On the other hand, when using MOF materials for photocatalytic degradation, pH should also be considered to affect the leaching of metal elements in MOF. Hu et al. [103] synthesized iron-based MIL-101(Fe) and established a MOF-photo-Fenton reaction system with multi-wavelength light + MIL-101(Fe) + H_2_O_2_ to eliminate TCEP. Under acidic conditions (pH = 3), the mass loss was only 1% after 60 min. However, with the increase in pH, the iron leakage phenomenon was aggravated. The system had good performance in degrading TCEP in a real wastewater matrix under acidic conditions. This was because the leaching of iron ions in MIL-101(Fe) under alkaline conditions was mainly due to its own decomposition. However, under acidic conditions, this was mainly attributed to the gradual oxidation of the surface iron-benzoic acid complex by •OH, resulting in the leaching of a small amount of Fe. There were 11 types of degradation products of TCEP detected in the photo-Fenton system, as well as other small molecule products such as chloroacetic acid, monochloroacetic acid, formic acid, and acetic acid. The main degradation pathways included cleavage, hydroxylation, carbonylation, and carboxylation. The presence of inorganic anions in the water had a potential inhibitory effect. Antonopoulou et al. [43] showed that inorganic anions would cover the available reactive sites of the photocatalyst for reaction with the organic substrate, forming a surrounding layer. In addition, some anions (e.g., NO_3_^−^, SO_4_^2−^) might raise the turbidity of the solution, which could cause the screening of UV radiation when applying photocatalytic treatments.

### 5.2. Adsorption Degradation Mechanism and Products

For the elucidation of the adsorption process, pseudo-first-order and pseudo-second-order adsorption models were generally used as expressed in the following (Equations (5) and (6)). Where t was time, *Q_t_* was the adsorption capacity at a particular time *t*, *Q_e_* was the equilibrium adsorption capacity, and *k*_1_ and *k*_2_ were the rate constants for the pseudo-first- and pseudo-second-order adsorptions, respectively. The correlation coefficient (*R*^2^) was used to describe which kinetic model the fitting results fit. The higher *k*_1_ and *k*_2_ of OPEs, the faster the adsorption, was confirmed [80,104].
(5)$$\mathrm{lg}(Q_{e}\;-\;Q_{t})=\mathrm{lg}Q_{e}\;-\frac{k_{1}}{t}$$
(6)$$\frac{t}{Q_{t}}=\frac{1}{{Q_{e}}^{2}\;\times \;k_{2}}+\frac{t}{\;Q_{e}}$$

Langmuir and Freundlich’s linear equations were used to fit the isotherm data as follows (Equations (7) and (8)) [73]:(7)$$\frac{1}{Q_{e}}=\frac{1}{c_{e}Q_{m}k_{1}}+\frac{t}{Q_{m}}$$
(8)$$\ln\;Q_{e}=\ln k_{F}+\frac{1}{n}\ln\;c_{e}$$
where *C_e_* was the equilibrium concentration of OPEs (ng·mL^−1^) in solution; *Q_e_* was the equilibrium adsorption capacity of OPEs per unit adsorbent (ng·g^−1^), *Q_m_* was the maximum adsorption capacity of the monolayer (ng·g^−1^); n was the surface heterogeneity factor. If *n* > 1, the adsorption was favorable, indicating that the adsorption capacity of the adsorbent would increase with a higher initial concentration of OPEs. *k_L_* (L·n g^−1^) and *k_F_* (n g^−1^/n·L^−1^/n·g^−1^) represented the affinity constants of the Langmuir and Freundlich equations, respectively. The correlation coefficient (*R*^2^) was used to describe the adsorption isotherm model. The Langmuir model presented a linear monolayer coverage sorption behavior over the sorbent. The monolayer sorption of OPEs on the adsorbents might occur via the hydrophobic interaction in the adsorption process. The Freundlich model was a nonlinear sorption model on a heterogeneous surface. The different adsorption interactions, such as hydrogen bonding and π-π interactions, could cause nonlinearity. According to Table 2, in the degradation of OPEs by adsorption, the adsorption kinetic models were mostly consistent with pseudo-second-order kinetic models, and the adsorption isothermal model was the Freundlich/Langmuir isothermal model.

The different effective adsorption sites on the adsorbents caused their different adsorption capacities. Figure 4 shows the reasonable conformation of the three OPEs and the electrostatic potential (ESP). The blue part of the ESP was the negative electrode region, which could be used as the adsorption site in the positive electrode region of the adsorbent. The red area was the positive zone, which acts opposite to the negative zone. OPEs were structurally similar to organophosphate pesticides and organophosphate nerve agents, such as the P–O bond [105]. Bazargan et al. [106] quantified the adsorption of the organophosphorus agent X(VX) on clean and hydroxylated (101) surfaces of anatase-type TiO_2_ with density functional theory (DFT) calculations. On the surface of pure anatase (101), VX phosphorylates the oxygen (O = P) site and induces the Ti⋯O = P formation. The spatial effect inhibited the adsorption of Ti⋯N and Ti⋯S direction-finding bonds on the VX nitrogen and sulfur sites. On hydroxylated (101) surfaces, adsorption also occurred via VX-phosphorylated oxygen sites but required the formation of surface adsorption-type hydrogen bonds. In addition, weak noncovalent interactions between the surface hydroxyl groups and the adsorbed nitrogen and sulfur atoms stabilize the VX/(101) complexes formed by adsorption through these secondary sites. TCEP, as the processing aid in production, could be detected in the soil environment along with wastewater discharge. The target was usually enriched by solid-phase extraction and used gas chromatography-mass spectrometry to determine the products. Zheng et al. [107] selected TnBP, TBEP, and TCEP, and their sorption properties on Pahokee peat soil with high organic matter, were investigated by solid-phase micro-extraction coupled with gas chromatography-mass spectrometry. The sorption kinetics of three types of OPEs on Pahokee peat soil were confirmed by the pseudo-second-order kinetic model. Thus, it could be concluded that the sorption proceeded with the boundary layer and intra-particle diffusion. The sorption equilibrium was fitted to the Langmuir isotherm model, in which TnBP exhibited the strongest affinity to Pahokee peat soil, followed by TBEP and TCEP. TnBP possessed the highest Kow and lowest water solubility. According to the results of the adsorption isothermal model, the adsorption properties of the three OPEs were all physical adsorption. Almerindo et al. [108] reported the transesterification reaction between OPEs and 1-propanol catalyzed by magnesium oxide. The characterization of the catalyst showed that the MgO was beneficial to the adsorption of methyl to oxygen and the transesterification reaction. The adsorption energies of 1-propanol and p-propanol on the surface of the MgO model were calculated by DFT, and the results confirmed that p-propanol had a stronger affinity for the catalyst. The reaction process should be that p-propanol and 1-propanol molecules were placed on the adjacent Mg^2+^ site, and the nucleophilic center and electrophilic center were about 2.4 Å apart. MgO also promoted the propanolysis rate of more representative triphosphate to 5 × 10^4^-fold. In most cases, the final transesterification products were trialkyl phosphates, which are structurally related to flame retardants.

Therefore, the adsorption and degradation processes of the OPEs machine could be summarized as follows: the widely used adsorbents rely on van der Waals forces, electrostatic interaction, and hydrogen bond interaction on OPEs adsorption; the selective adsorbents were designed to degrade the corresponding groups quickly by complexation. Moreover, most adsorbents could be reused after adsorption and desorption, and the adsorption performance of the adsorbents after recycling had little effect. The high-efficiency adsorbents had the advantages of large specific surface area, large pore size, strong mechanical properties, and recycling [109]. The adsorption performance was affected by pH, temperature, substrate concentration, and inorganic anion. OPEs were adsorbed by an adsorbent and degraded by water, and the toxicity of the degraded intermediates decreased.

### 5.3. Biological Degradation Mechanism and Products

In general, the effluent of WWTPs that treated production and domestic sewage was considered to be the major source of OPEs to water environment pollution [110]. The adsorption of the waste-activated sludge was an important factor in the removal of OPEs in the treatment process, so the sludge enriched a high level of OPEs. The degradation of OPE compounds by microorganisms was mainly through the hydrolysis of phosphate ester groups by enzymes such as organophosphorus hydrolase or phosphotriesterase. Pang et al. [111] studied the degradation of OPEs during aerobic composting and anaerobic digestion of sewage sludge. It has been found that the addition of pig manure or sawdust promotes the degradation of OPEs. This was due to the fact that in addition to the adsorption of sludge, the pig manure might produce xanthobacteria, bacillus, alcaligene, pseudomonas, bacillus megaterium, and sphingosine bacteria during aerobic composting and anaerobic digestion, which were related to the degradation of organophosphorus compounds, increasing the removal rate of OPEs by 10–50%. Other bacteria might also be able to degrade pollutants after long-term domestication. The space effect might be another reason for the poor degradation of chlorinated OPEs because it hinders the attack of hydrolases on pollutants. In addition, the biotoxicity of pollutants might inhibit the hydrolytic release of enzymes, which might lead to a decrease in the biodegradation efficiency of chlorinated OPEs [112].

Chen et al. [10] developed a sensitive quantitative method used to analyze several OPEs in herring gull eggs in Lake Huron. The method was based on a simple two-step sample extraction, and then the TCPP, TCEP, and TBEP in herring gull eggs were successfully detected by liquid chromatography-electrospray ionization (+)-tandem mass spectrometry, which indicated bioaccumulation potential. Furthermore, the study showed that triester OPEs could be metabolized into phosphoric acid diesters and monoesters. TEHP metabolism to di(2-ethylhexyl) phosphate (DEHP) and mono(2-ethylhexyl) phosphate (MEHP) [18], and tris(1,3-dichloro-2-propyl) phosphate (TDCPP) metabolismed to di(1,3-dichloro-2-propyl) phosphate.

Triester and diester OPEs in aquatic environments mainly came from municipal wastewater from WWTPs [113]. Li et al. [114] found that, among the 13 types of OPEs, TEP content in Chinese river water was the highest (average 1.48 × 10^3^ ng/L), followed by TCEP (average 190 ng/L) and TCIPP (average 158 ng/L). Three organophosphorus diesters (DNBP, DPHP, and BDCIPP) were potential organophosphorus triester degradation products. Hou et al. [91] studied the feasibility of using the BES system to degrade TPHP and several non-chlorinated OPEs after a certain acclimation period. The study found that TPHP could be degraded into DPHP, hydroxy-triphenyl phosphate, and three by-products. The main products of ultrasonic treatment TCEP were monoester and diester. Zhu et al. [44] reported two enriched cultures containing dehalococcoides that transformed TCPP and TCEP within 10 days. The results deduced that TCEP might be converted to di (2-chloroethyl) phosphate and ethylene based on the identification of the conversion products and deuteration experiments by radical mechanism followed by C–O bond cleavage. Ethylene was then reduced to ethane. Similarly, TCPP was converted to bis (1-chloro-2-propyl) phosphate and propene. The 16SrRNA gene amplification sequencing and quantitative PCR analysis showed that dehalogenases bacteria were the main contributors to TCEP and TCPP transformation. The degradation process of two chlorinated OPEs is shown in Figure 5A,B. To date, no efficient sample cleaning and analysis method has been developed to synchronously identify all possible OPEs and their conversion products due to their numerous congeners and differing physicochemical properties. Screening analysis based on high-resolution mass spectrometry such as TOF-MS might be a useful tool for further research on OPEs [115].

## 6. Conclusions and Outlook

In this review, we summarized the three commonly used methods to degrade OPEs and gained some key insights from a series of studies. These basic understandings could enrich our understanding of OPE degradation:(1)In photocatalysis, •OH played a major role, but the common problem was that the composite of photogenerated electrons and holes affected the performance of the photocatalyst. Therefore, improving the photoresponse of the photocatalyst was an important challenge to improving the photocatalytic degradation of OPEs;(2)The degradation mechanism of the adsorption method mainly involved van der Waals forces, hydrophobic interaction, electrostatic attraction, the hydrogen bond, and π-π interaction. The biological method had a long treatment cycle and the lowest treatment efficiency, so there were few studies on this aspect;(3)In general, all three methods were affected by catalysts/adsorbents activity, substrate properties, temperature, pH, inorganic anions, and HA. The degradation rate of OPEs was related to their structure; alkyl-OPEs were easier to degrade than halogenated-OPEs and aryl-OPEs, and triester OPEs could be degraded to phosphoric acid diesters and monoesters. Toxicity analysis showed that the toxicity of the degradation intermediates produced by the three methods decreased.

So far, there have been studies on the distribution and concentration of OPEs in air, water, soil, animals, and plants, as well as different degradation methods on OPEs. In general, the research on OPE degradation methods had a great prospect, but there were also some challenges and breakthroughs to be made. Firstly, for the photocatalytic method, on the one hand, the methods to improve the photocatalytic performance of materials had shortcomings such as high cost and complex modification methods. On the other hand, the problem of new pollutants leaching might occur if the photocatalyst was not recovered in time. Secondly, for the adsorption method, the desorption phenomenon might occur if the adsorbent was not recovered in time, leading to a reduction in degradation efficiency. Finally, for the biological methods, most research processing times were too long, and the degradation efficiency was not as good as the first two methods. Thus, it was necessary to develop more materials with good recycling properties for practical application in future wastewater treatment. In addition, the actual water environment was complex, but the most current research was limited to the treatment of a single OPE. It was necessary to study the complex wastewater environment mixed with multiple pollutants in the future. It was also urgent to further study whether OPEs may generate more/fewer toxic metabolites or by-products (such as organophosphorus diesters and hydroxylated metabolites) after photodegradation, biotransformation, and hydrolysis processes under environmental conditions. In the future, new ways of combining multiple technologies for efficiently degrading OPEs should be explored in combination with multiple factors. So, the summary of different methods in this paper provided some understanding for the future study of OPE degradation.

## Figures and Tables

**Figure 1 molecules-28-02983-f001:**
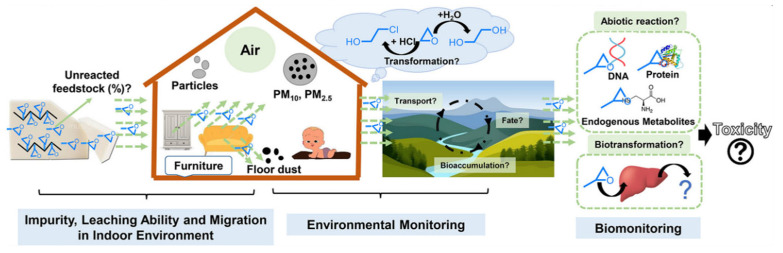
Knowledge gaps and future studies for reactive flame retardants [23].

**Figure 2 molecules-28-02983-f002:**
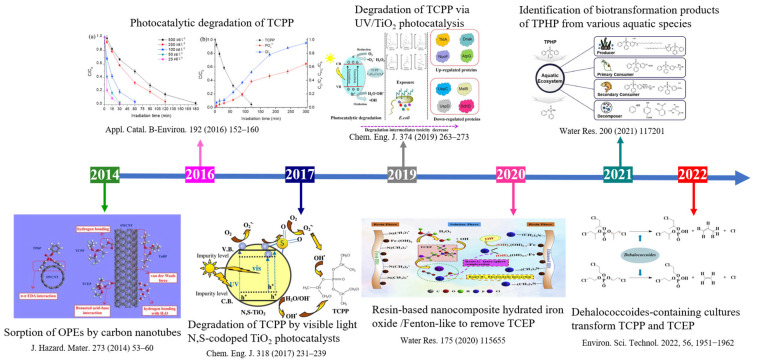
The development timeline of treatment methods for OPEs degradation. [38] Chem. Eng. J. 374 (2019) 263–273; [39] Chem. Eng. J. 318 (2017) 231–239; [40] J. Hazard. Mater. 273 (2014) 53–60; [41] Water Res. 175 (2020) 115655; [42] Water Res. 200 (2021) 117201; [43] Appl. Catal. B-Environ. 192 (2016) 152–160; [44] Environ. Sci. Technol. 2022, 56, 1951–1962.

**Figure 3 molecules-28-02983-f003:**
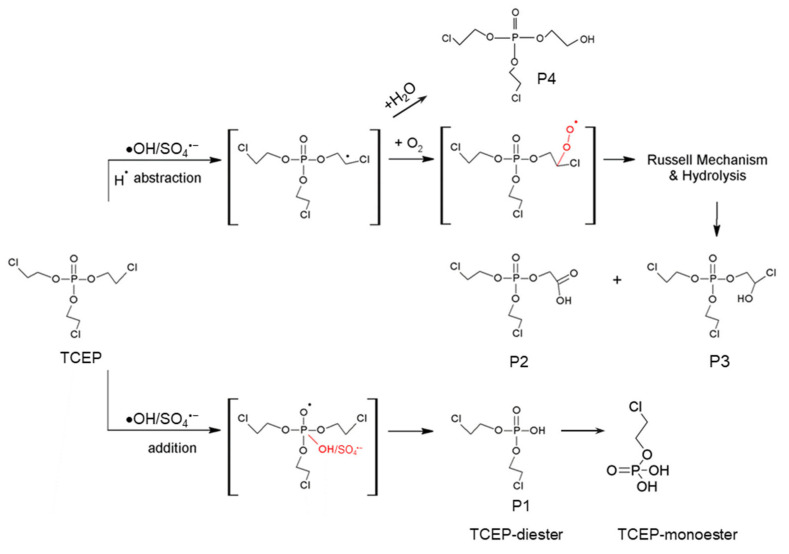
The degradation pathways of TCEP radical oxidation [27,101].

**Figure 4 molecules-28-02983-f004:**
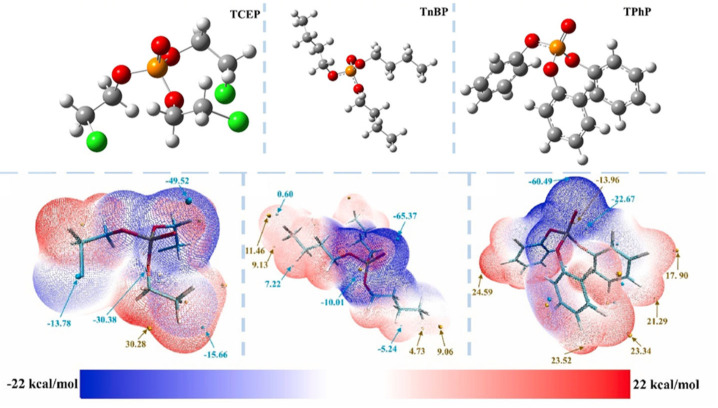
The reasonable conformation of the three OPEs and the ESP (Green represented Cl atom, orange represented P atom, and red represented O atoms in conformations) [109].

**Figure 5 molecules-28-02983-f005:**
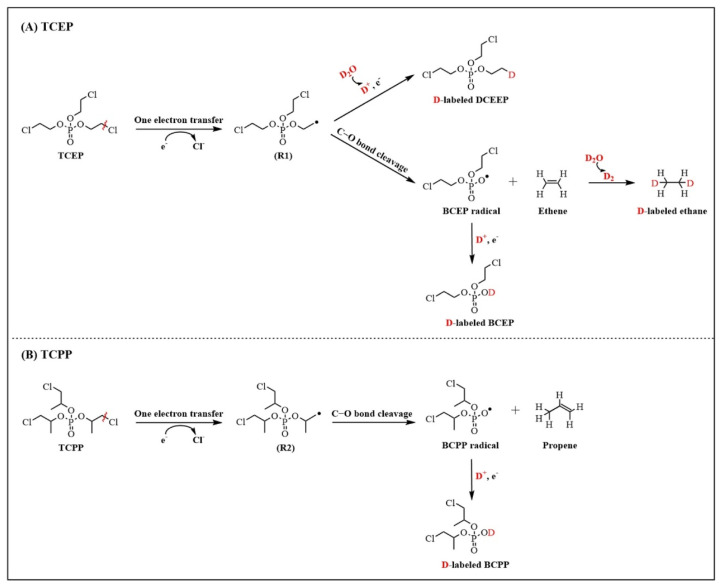
Schematic diagram showing potential mechanisms for the transformation of TCEP to ethene/BCEP (**A**) and of TCPP to propene/BCPP (**B**) in media prepared with D_2_O [44].

**Table 1 molecules-28-02983-t001:** Degradation of OPEs by photocatalysis.

Targets	Chemical Structures	Method	Light	Mechanism	Ref.
TBEP	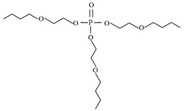	TiO_2_/V_2_O_5_	Vis	•OH, carboxylation, hydroxylation, dechlorination	[69]
		(N, F-doped)-TiO_2_/V_2_O_5_			
		N-doped-SrTiO_3_			
TCEP	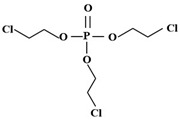	185 + 254 nm	UV	•OH attacked the center of phosphate and terminal Cl^−^	[54]
TCEP	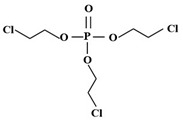	TiO_2_ + 350 nm	UV	•OH, initially oxidized to diesters, then to monoesters, and finally to phosphates	[27]
TCPP	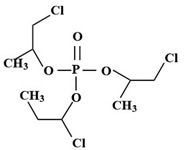	UV/H_2_O_2_	UV	•OH	[51]
TCPP	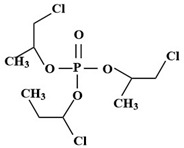	UV/TiO_2_	UV	•OH	[38]
TCEP	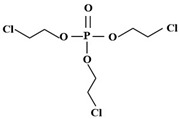	MIL-101(Fe) + PMS	420 nm	SO_4_^•−^	[57]
TCPP	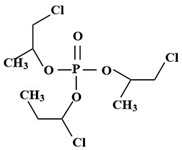	MIL-88A + H_2_O_2_	Vis	•OH	[58]
TCPP	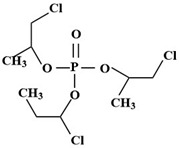	MIL-88B-NH_2_ + H_2_O_2_	Vis	•OH, carboxylation, hydroxylation, dechlorination	[61]
TCEP	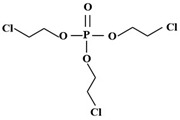	GO@MIL-101(Fe)	Vis	•OH, hydroxylation, carbonylation, carboxylation	[62]
TCPP	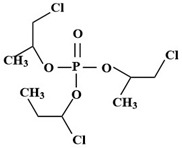	N/N, S doped-TiO_2_	UV-vis	Under the simulated sunlight was •OH Under visible light was ROS	[39]
TBEP	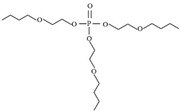	V_2_O_5_/TiO_2_ N-doped-SrTiO_3_	UV-vis	•OH, multiple hydroxylation, and oxidation	[69]

**Table 2 molecules-28-02983-t002:** Degradation of OPEs by adsorption.

Targets	Kinetics	Isotherms	Method	Mechanism	Ref.
TEP, TCEP, TCPP, TBP, TPhP	Pseudo-second	Langmuir/Freundlich	Four types AC (GAC) (PAC) (R-GAC) (O-GAC)	Hydrophobic effect, π-π interactions, electrostatic attraction, hydrogen bonding	[70]
TCEP, TCPP, TnBP, TBEP, TPhP	-	Dubinin–Ashtakhov	Four types carbon nanotubes (MWCNTs), (SWCNTs), (O-MWCNTs) (O-SWCNTs)	-	[40]
TCPP	Pseudo-second	Langmuir/Freundlich	PC/Nano-Fe_3_O_4_ composites	van der Waals forces, EDA interaction, hydrogen bonding	[71]
TCEP, TCPP, TBP, TPhP, TPPO	Pseudo-second	Langmuir	2.5 mg resins (XAD4 and XAD7hP)	Monolayer adsorption, electrostatic and hydrogen bond interactions	[72]
TCEP	Pseudo-first	Langmuir/Freundlich	HD1/H_2_O_2_	electrostatic adsorption and inner layer complexation	[41]
TnBP, TCEP	Pseudo-first order/pseudo-second	Langmuir/Freundlich	PE/PVC	pore-filling and monolayer coverage	[73]

## Data Availability

No data were used for the research described in the article.

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
