# Peer review of "Organophosphate Esters (OPEs) Flame Retardants in Water: A Review of Photocatalysis, Adsorption, and Biological Degradation"

_molecules, 2023, doi:10.3390/molecules28072983_

Round 1

Reviewer 1 Report

Review Report

Manuscript ID: molecules-2304258

Title: Organophosphate esters (OPEs) flame retardants in water: A review of photocatalysis, adsorption and biological degradation

 In this manuscript

 This paper reviewed methods to degrade OPEs, the photocatalytic method, the adsorption method, and the biological method (relying on enzymolysis and hydrolysis). The authors have successfully pointed out the advantages and disadvantages of the high removal efficiency of OPEs. Furthermore, the degradation efficiency of OPEs, mechanism, and conversion products of all three methods was deliberated. The development perspective and future tasks of OPEs degradation technology were also elaborated.

I recommend that the Editorial office consider this manuscript for publication after minor revision.

Reviewer’s Suggestions

Line 34 "And the current studies had detected …..", maybe it’s better not to begin a sentence with And.

Line 67 Which fish and larvae?

Line 78 "soluble in organic solvents" such as?  

Line 229 Show reference to support Eq. 2-4.

Line 230 "In addition, there were studies had found that comparing …" This sentence sounds wrong, it has to be rewritten.

Line 245 "For example, M et al. [55] ", you mean Antonopoulou et al.

Author Response

  1. Line 34 "And the current studies had detected …..", maybe it’s better not to begin a sentence with And.

Answers 1: Thanks for your comments and suggestions. Your suggestions and comments have greatly helped us to improve the quality of our manuscript. We revised the sentence to “The current studies had detected the presence of these substances in water, air, soil, fish, human body and other ecosystem environments.”

  1. Line 67 Which fish and larvae?

Answers 2: Thanks for your comments and suggestions. Your suggestions and comments have greatly helped us to improve the quality of our manuscript. The correct word should be killifish and zebrafish larvae. We have modified the paper and marked it in red.

  1. Line 78 "soluble in organic solvents" such as?  

Answers 3: Thanks for your comments and suggestions. Organophosphate esters are organic matters, and most tri-OPEs have strong hydrophobicity. Organophosphate esters can dissolve in acetone, ethanol, benzene and other general organic solvents.

  1. Line 229 Show reference to support Eq. 2-4.

Answers 4: Thanks for your comments and suggestions. Your suggestions and comments have greatly helped us to improve the quality of our manuscript. The formula reference is from the reference: “ Li, X.Y.; Jie, B.R.; Lin, H.D.; Deng, Z.P.; Qian, J.Y.; Yang, Y.Q.; Zhang, X.D. Application of sulfate radicals-based advanced oxidation technology in degradation of trace organic contaminants (TrOCs): Recent advances and prospects. J. Environ. Manage. 2022, 308, 114664.” We have added it in paper and marked it in red as [56].

  1. Line 230 "In addition, there were studies had found that comparing …" This sentence sounds wrong, it has to be rewritten.

Answers 5: Thanks for your comments and suggestions. Your suggestions and comments have greatly helped us to improve the quality of our manuscript. We have revised the content as following.

Line 230: In addition, in the advanced oxidation methods, sulfate free radical S-AOPs produced SO4•- with stronger oxidation potential (2.6 V) in a wider pH range, and produced fewer toxic by-products than the •OH [53].

  1. Line 245 "For example, M et al. [55] ", you mean Antonopoulou et al.

Answers 6: Thanks for your comments and suggestions. Your suggestions and comments have greatly helped us to improve the quality of our manuscript. We have modified the paper and marked it in red.

Reviewer 2 Report

The review manuscript entitled “Organophosphate esters (OPEs) flame retardants in water: A review of 2 photocatalysis, adsorption and biological degradation” by Zhang and Liu et al summarized the photocatalysis method, the adsorption method with wide applicability, and the biological method mainly relying on enzymolysis and hydrolysis to degrade OPEs in water. I think this is a significant topic and suitable for publication in Molecules. As a review, there are some issues that need to be addressed.

1. It is recommended to add the selected chemical structure formulas of the classical OPEs in Tables 1&2.
2. The authors claimed that "Toxicity analysis showed that the toxicity of the degradation products by the three methods was decreased." It would have been better if this could have been quantified further, as the goal was not a decrease in toxicity but the absence of toxicity.
3. I note that the authors have cited a number of metal oxides and metal organic frameworks, in fact the potential of  covalent organic frameworks and their composites cannot be ignored, and suggest adding relevant discussions and citations(Molecules 2022, 27, 8002.).

Author Response

  1. It is recommended to add the selected chemical structure formulas of the classical OPEs in Tables 1&2.

Answers 1: Thanks for your comments and suggestions. Your suggestions and comments have greatly helped us to improve the quality of our manuscript. We have added the chemical structure formulas of classical OPEs to Table 1. Due to the degradation of multiple OPEs in Table 2, and the duplication of the main degraded OPEs with the structure in Table 1, major changes have been made to Table 1. The revised content as following.

  1. The authors claimed that "Toxicity analysis showed that the toxicity of the degradation products by the three methods was decreased." It would have been better if this could have been quantified further, as the goal was not a decrease in toxicity but the absence of toxicity.

Answers 2: Thanks for your comments and suggestions. Your suggestions and comments have greatly helped us to improve the quality of our manuscript. As an organic substance, the final degradation products of OPEs include PO43-, CO2, and H2O. For example, the formula shows:

The toxicity analysis here mainly focuses on various intermediate products. My modification result for this sentence is: Toxicity analysis showed that the toxicity of the degradation intermediates by the three methods was decreased.

  1. I note that the authors have cited a number of metal oxides and metal organic frameworks, in fact the potential of covalent organic frameworks and their composites cannot be ignored, and suggest adding relevant discussions and citations (Molecules 2022, 27, 8002.).

Answers 3: Thanks for your comments and suggestions. Your suggestions and comments have greatly helped us to improve the quality of our manuscript. We have added relevant references in the revised article and marked them in red 48.